# Structure of cellular ESCRT-III spirals and their relationship to HIV budding

**Anil G Cashikar[†], Soomin Shim[†], Robyn Roth, Michael R Maldazys, John E Heuser, Phyllis I Hanson\***

Department of Cell Biology and Physiology, Washington University School of Medicine, St. Louis, United States

**Abstract** The ESCRT machinery along with the AAA+ ATPase Vps4 drive membrane scission for trafficking into multivesicular bodies in the endocytic pathway and for the topologically related processes of viral budding and cytokinesis, but how they accomplish this remains unclear. Using deep-etch electron microscopy, we find that endogenous ESCRT-III filaments stabilized by depleting cells of Vps4 create uniform membrane-deforming conical spirals which are assemblies of specific ESCRT-III heteropolymers. To explore functional roles for ESCRT-III filaments, we examine HIV-1 Gag-mediated budding of virus-like particles and find that depleting Vps4 traps ESCRT-III filaments around nascent Gag assemblies. Interpolating between the observed structures suggests a new role for Vps4 in separating ESCRT-III from Gag or other cargo to allow centripetal growth of a neck constricting ESCRT-III spiral.

**\*For correspondence:** phanson22@WUSTL.EDU

[†]These authors contributed equally to this work

**Competing interests:** The authors declare that no competing interests exist.

**Reviewing editor**: Wesley Sundquist, University of Utah, United States

## Introduction

Vesicle formation requires machinery to identify cargo, deform the membrane and release a vesicle. How cytoplasmic proteins cooperate to form vesicles that mediate transport between organelles is well defined. Less well understood is how cytoplasmic proteins drive the topologically reverse process of creating intralumenal vesicles (ILVs) in multivesicular bodies (MVBs). Studies over the past decade have shown that a set of over 30 proteins are responsible for ILV formation and are also required for the topologically similar processes of viral budding and cytokinesis (*Hurley, 2010*; *Hanson and Cashikar, 2012*; *Sundquist and Krausslich, 2012*; *Henne et al., 2013*; *McCullough et al., 2013*). Most are components of ESCRT (endosomal sorting complex required for transport) complexes. The multisubunit ESCRT-0, -I, and -II complexes are thought to recognize cargo and initiate vesicle formation, while ESCRT-III proteins and the AAA+ ATPase Vps4 that acts on them promote membrane remodeling and scission both on MVBs and the plasma membrane. A subject of much current interest is how, exactly, ESCRT-III and Vps4 drive these events.

Seven ESCRT-III proteins in yeast and 12 in humans are structurally similar to each other but are not redundant for pathway function (*Morita et al., 2010*, *2011*; *Peel et al., 2011*). ESCRT-III proteins cycle between an inactive or closed monomeric state in the cytoplasm and an activated or open state in which they polymerize into filaments on the membrane (*Babst et al., 2002*; *Lin et al., 2005*; *Zamborlini et al., 2006*; *Shim et al., 2007*; *Hanson et al., 2008*). ESCRT-II initiates ESCRT-III assembly on the MVB by binding to Vps20/CHMP6 (*Babst et al., 2002*; *Teis et al., 2010*) while virally encoded sequence motifs bind ESCRT-I and/or Alix which in turn promote ESCRT-III assembly during viral budding (*von Schwedler et al., 2003*; *Strack et al., 2003*). Once nucleated, ESCRT-III polymers grow by recruiting other subunits (*Babst et al., 2002*; *Shim et al., 2007*; *Teis et al., 2008*; *Saksena et al., 2009*). Reconstituting ILV formation on giant unilamellar vesicles established that ESCRT-III proteins alone can promote vesicle release (*Wollert et al., 2009*; *Wollert and Hurley, 2010*) although studies in yeast suggest that they normally do this in cooperation with Vps4 (*Nickerson et al., 2010*;

**eLife digest** Cells contain compartments called organelles that are enclosed within membranes similar to the plasma membrane that surrounds the cell itself. Cells police the proteins on their membranes and move old or damaged proteins into a type of organelle called an endosome. This involves the membrane folding in on itself to form a multivesicular body.

The multivesicular bodies deliver their contents to organelles called lysosomes where the old proteins are destroyed. Although it is known that over 30 proteins are involved in the formation of multivesicular bodies, many aspects of how they operate are not well understood. Moreover, disruptions to this process contribute to a large number of diseases including forms of cancer and neurodegeneration. Importantly, the same proteins are hijacked by viruses such as HIV to help them escape from the cells they have infected.

Most of the proteins involved in forming multivesicular bodies are part of the ESCRT (Endosomal Sorting Complex Required for Transport) system of proteins. A special set of these proteins—ESCRT-III—is thought to cut the membrane to release vesicles and viruses, as well as helping the membrane to deform.

Previously, researchers have been unsure how the ESCRT-III complex works because it has a short lifespan and is too small to see on cellular membranes using standard techniques. Now Cashikar, Shim et al. have used a technique called deep-etch electron microscopy in combination with gene knockdown strategies to reveal the structure of the ESCRT-III complex inside cells.

A protein called Vps4 is known to recycle ESCRT-III complexes, so Cashikar, Shim et al. studied cells in which the levels of Vps4 had been depleted in order to increase the lifespan of ESCRT-III complexes. In these cells filaments made of ESCRT-III complexes tended to form conical spirals that matched the size and shape of the vesicles and viruses ESCRT-III is thought to produce. ESCRT-III filaments also accumulated as rings around the molecules destined for incorporation into a vesicle or virus. This indicated a new role for Vps4: it separates ESCRT-III from the contents of the vesicle, leaving it free to form a spiral that drives release of the vesicle or virus from the cell.

The next challenge will be to test the predictions of this model using techniques that can capture individual vesicle formation events in real time. Understanding the function of ESCRT-III in greater detail may suggest ways to manipulate this pathway to limit the replication of viruses or the degradation of membrane proteins.

*Adell et al., 2014*). In addition to Vps20/CHMP6 as a nucleator, specific roles attributed to individual ESCRT-III proteins include cargo confinement by Snf7/CHMP4 (*Teis et al., 2008*), polymer capping by Vps24/CHMP3 (*Saksena et al., 2009*), and Vps4 recruitment by Vps2/CHMP2 and Did2/CHMP1 (*Lata et al., 2008*; *Saksena et al., 2009*; *Davies et al., 2010*; *Adell et al., 2014*). ESCRT-III polymers are remodeled and disassembled by the AAA ATPase Vps4 before (*Nickerson et al., 2010*; *Baumgartel et al., 2011*; *Jouvenet et al., 2011*; *Adell et al., 2014*) and/or after (*Wollert et al., 2009*; *Wollert and Hurley, 2010*) membrane scission, returning individual subunits to their closed state in the cytoplasm.

Structural studies demonstrate that ESCRT-III proteins each consist of a ~7 nm helical hairpin (α1–α2) stabilized and regulated by four or more short helices (α3–α6) (*Muziol et al., 2006*; *Bajorek et al., 2009*; *Xiao et al., 2009*; *Martinelli et al., 2012*). α5–α6 and surrounding sequences are responsible for autoinhibition and thereby regulate membrane binding and polymer assembly (*Zamborlini et al., 2006*; *Shim et al., 2007*; *Bajorek et al., 2009*). In their active or open state, ESCRT-III proteins polymerize in vitro and in vivo using a number of different protein–protein interfaces (*Muziol et al., 2006*; *Ghazi-Tabatabai et al., 2008*; *Bajorek et al., 2009*; *Xiao et al., 2009*; *Morita et al., 2011*). Snf7/CHMP4 is the most abundant ESCRT-III protein, and forms membrane-associated polymers in vitro and in transfected cells (*Hanson et al., 2008*; *Teis et al., 2008*; *Fyfe et al., 2011*; *Henne et al., 2012*). Electron microscopy (EM) shows that Snf7/CHMP4 polymers are curved filaments <5 nm in diameter (*Ghazi-Tabatabai et al., 2008*; *Hanson et al., 2008*; *Pires et al., 2009*; *Henne et al., 2012*). Other ESCRT-III proteins shown to assemble in vitro into a variety of mostly tubular structures include CHMP2A and CHMP3 (*Lata et al., 2008*; *Bajorek et al., 2009*; *Dobro et al., 2013*; *Effantin et al., 2013*), CHMP1B (*Bajorek et al., 2009*; *Dobro et al., 2013*), CHMP2B (*Bodon et al., 2011*), and Ist1 (*Bajorek et al., 2009*; *Dobro et al., 2013*). Finally, ~17 nm filaments on the membrane in intercellular

bridges before cytokinesis disappear when CHMP2A is missing, suggesting that they too might represent a form of ESCRT-III polymer (*Guizetti et al., 2011*). These many views of ESCRT-III have led to proposals for how it and Vps4 contribute to membrane scission during ILV formation, viral budding, and cytokinesis (*Fabbro et al., 2005*; *Lenz et al., 2009*; *Saksena et al., 2009*; *Wollert et al., 2009*; *Elia et al., 2012*; *Henne et al., 2013*; *McCullough et al., 2013*). However, much remains to be learned about the assembly and functional roles of different types of ESCRT-III polymers in their own right and in conjunction with Vps4.

For insight into ESCRT-III polymer structure in a physiological setting, we examined endogenous machinery in cultured mammalian cells by deep-etch EM. We find that ESCRT-III heteropolymers stabilized by depleting Vps4A & B are membrane-attached filaments that frequently spiral to delineate and fill circular domains ~110 nm in diameter. Many of these spirals induce conical deformations directed away from the cytoplasm. Filaments built from transfected proteins show that coassembly of two ESCRT-III proteins (in this case Snf7/CHMP4A and CHMP2A) is required to create membrane deforming ESCRT-III spirals. To define the relationship between ESCRT-III filaments and cargo, we studied HIV-1 Gag-driven virus-like-particle (VLP) assembly. Notably, in cells depleted of Vps4, we find that Gag assemblies are often encircled by ESCRT-III filaments. This suggests a previously unappreciated role for Vps4 in remodeling ESCRT-III around cargo-containing membrane domains in addition to its more canonical role in recycling ESCRT-III subunits.

## Results and discussion

### Depleting Vps4 in mammalian cells traps ESCRT-III polymers on membranes

Endogenous ESCRT-III proteins including CHMP4A and CHMP2B are diffusely localized in cells with little steady-state concentration on endosomes despite their known role in lumenal vesicle formation (*Figure 1A*, left panels). This is not surprising given measurements in live cells showing that both they and Vps4 are present for a few minutes or less at their site of action during viral particle release (*Baumgartel et al., 2011*; *Jouvenet et al., 2011*) or cytokinetic abscission (*Elia et al., 2012*). To visualize ESCRT-III, we therefore set out to increase the lifetime of membrane associated polymers taking advantage of the known role for Vps4 in mediating disassembly and recycling of ESCRT-III (*Babst et al., 1998*; *Lin et al., 2005*). Depleting Vps4A & B, the two Vps4 proteins present in human cells, by RNAi (*Figure 1B*) caused ESCRT-III proteins to redistribute onto puncta localized throughout the cell (*Figure 1A*, middle panels). Sedimentation of detergent solubilized cell extracts further demonstrated that a significant fraction of ESCRT-III proteins accumulated in the detergent insoluble fraction characteristic of polymerized ESCRT-III (*Figure 1C*; *Shim et al., 2007*).

Deep-etch EM is particularly well suited to examining the morphology of the plasma membrane (*Heuser, 2000*), and although many ESCRT-III puncta were on endosome like structures throughout the cytoplasm, a subset appeared at the edge of the cell and remained associated with unroofed plasma membranes (*Figure 1A*, right panels). Their presence could reflect a role for ESCRT-III in creating outwardly-directed vesicles known as ectosomes that bud from the plasma membrane (*Stein and Luzio, 1991*; *Nabhan et al., 2012*; *Choudhuri et al., 2014*; *Jimenez et al., 2014*). Alternatively, ESCRT-III puncta could be transferred to the plasma membrane by MVB exocytosis (*Thery, 2011*). Finally, ESCRT-III may have some other, potentially structural, role on the plasma membrane. Importantly, the accessibility of the plasma membrane to deep-etch EM provided us with a starting point for studying the structure of cellular ESCRT-III.

### Endogenous ESCRT-III polymers form conical spirals

Deep-etch EM of plasma membranes from cultured cells depleted of Vps4A & B revealed unique filamentous assemblies intermingled among otherwise typical cytoskeletal structures, clathrin lattices, and caveolae (*Figure 2A*). The new filaments were attached to the membrane and most clearly recognized in discrete spirals that often deformed the membrane into conical protrusions. Because these filaments resemble previously described ESCRT-III polymers, we used antibodies recognizing several ESCRT-III proteins and found that gold particles marking each of the proteins examined (CHMP6, CHMP2B, and CHMP1B) bound on or near filaments and filament spirals but not elsewhere on membranes from cells depleted of Vps4A & B (*Figure 2B*) or on membranes from control HeLa cells (not shown). Fortuitous immunodecoration inside broken but not fully unroofed Vps4-depleted cells

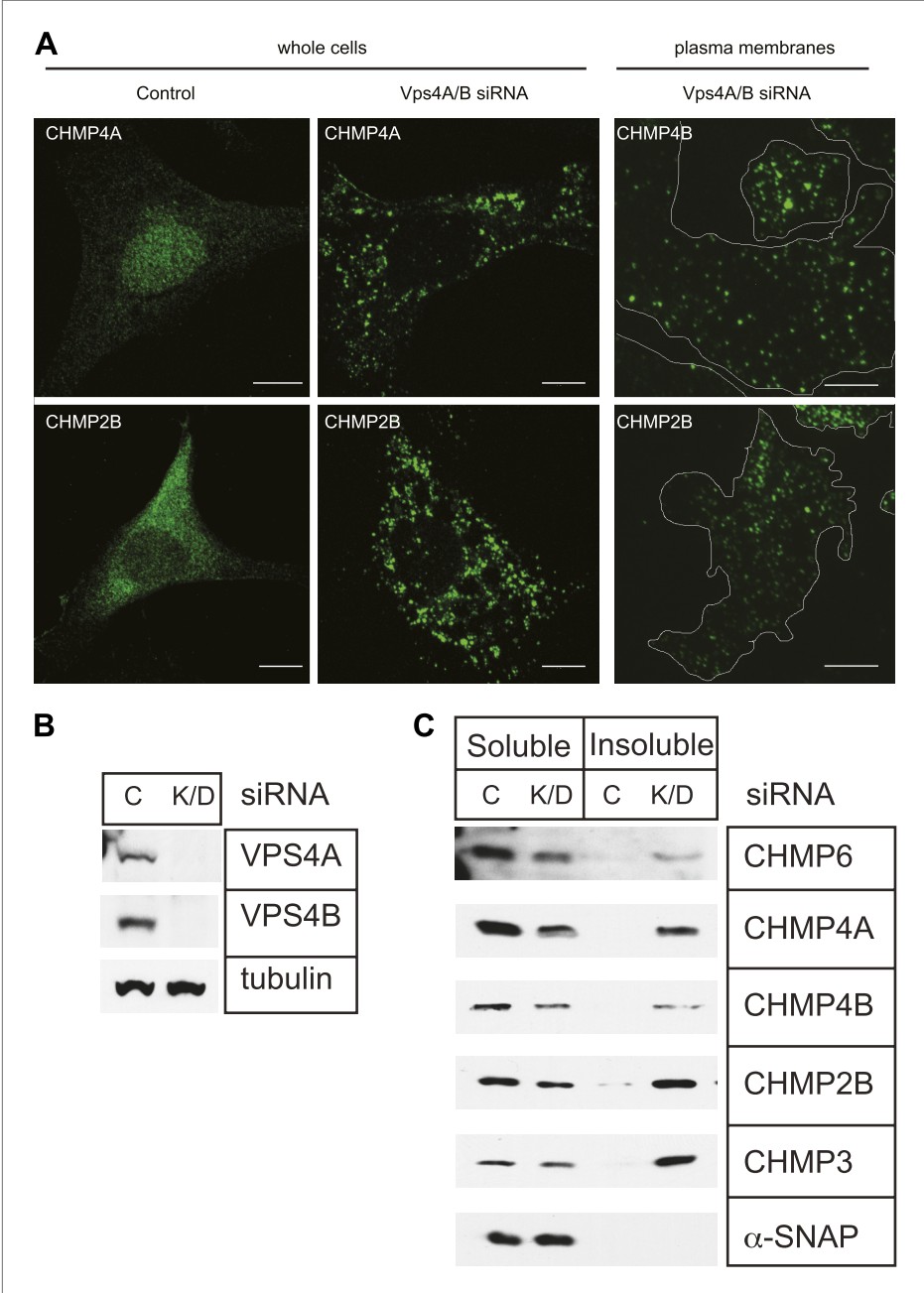

**Figure 1**. Effects of depleting Vps4 on ESCRT-III. (**A**) Localization of indicated ESCRT-III protein in HeLa cells untreated or treated with Vps4A & Vps4B specific siRNA for ~60 hr. Left and middle panels show maximum intensity projections from confocal z-series through the cells. Right panel shows immunostaining of unroofed plasma membranes from HEK293T cells treated with Vps4A & Vps4B siRNA. Scale bars represent 10 μm. (**B**) Representative immunoblots comparing lysates from cells untreated or treated with siRNA targeting Vps4A & Vps4B. (**C**) Immunoblots of detergent soluble (left) and insoluble (right) material from control or Vps4A & Vps4B siRNA-treated HeLa cells.

showed that eversions containing ESCRT-III were also present along the cell's top surface (*Figure 2— figure supplement 1*). The apparent distribution of different ESCRT-III proteins on the filament spirals was similar (*Figure 2B* and not shown), although generally predominant labeling near the perimeter may reflect limited epitope accessibility toward the center of the spirals. This immunodecoration argues against unique positioning of particular ESCRT-III proteins along the filaments, although higher

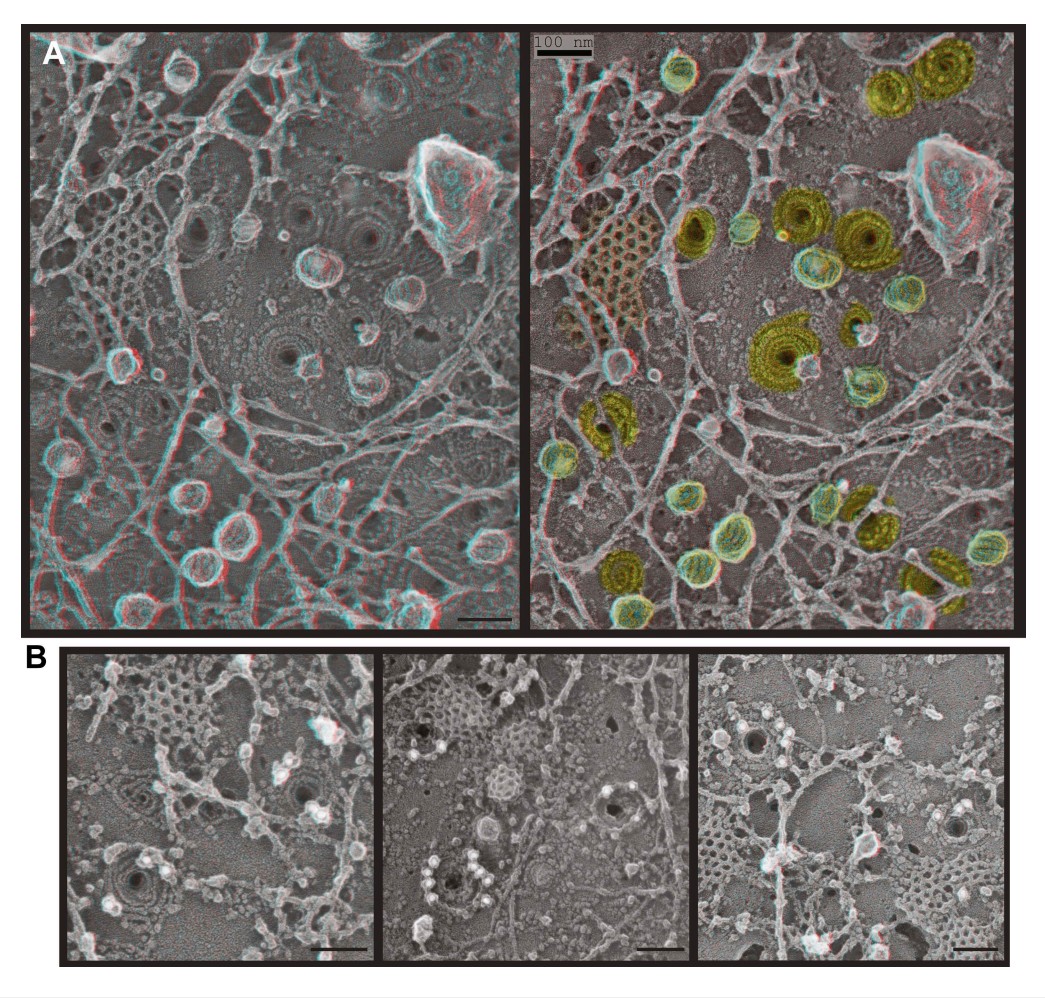

**Figure 2**. ESCRT-III filaments form conical spirals on the plasma membrane of cells depleted of Vps4A & Vps4B. (**A**) Survey view of the cytoplasmic surface of the plasma membrane from a HeLa cell unroofed ~60 hr after transfection with siRNA targeting Vps4A & Vps4B. Pseudocoloring shows clathrin (orange), caveolae (green), and ESCRT-III spirals (yellow). Similar filament spirals were seen in siRNA treated HEK293T, U2OS, and MCF-7 cells (not shown). (**B**) Immunodecoration of ESCRT-III proteins in spirals on Vps4-depleted HeLa cell plasma membranes. Antibodies recognizing CHMP6 (left), CHMP2B (middle), and CHMP1B (right) detected with 18 nm gold that appears white in these contrast reversed EM images. Use view glasses for 3D structure in both panels (left eye = red). Scale bars represent 100 nm.

The following figure supplements are available for figure 2:

**Figure supplement 1**. Immunodecoration of ESCRT-III proteins beneath protrusions from the cell surface.

resolution studies and additional antibodies will be required to determine exactly where different ESCRT-III proteins are present in each spiral.

The most striking and unexpected characteristic of these stabilized ESCRT-III filaments was their frequent organization into spirals (*Figure 3A,B*). Other patterns including partial or complete rings and meandering filaments were also seen but were less common. The overall diameter of the spiral assemblies was 108 ± 30 nm (*Figure 3C*), with an average of four 360° turns in each spiral. The shape of the spirals on the membrane varied, ranging from clearly conical to flat. The total length of the filament in each spiral was 660 ± 350 nm. While Vps4-driven remodeling may normally keep filaments from reaching this length (*Teis et al., 2008*), these images set an upper limit on the size and shape of endogenous ESCRT-III polymer that is smaller and more uniform than that of the extensive polymers seen in vitro (*Lata et al., 2008*; *Bajorek et al., 2009*; *Henne et al., 2012*) or in cells

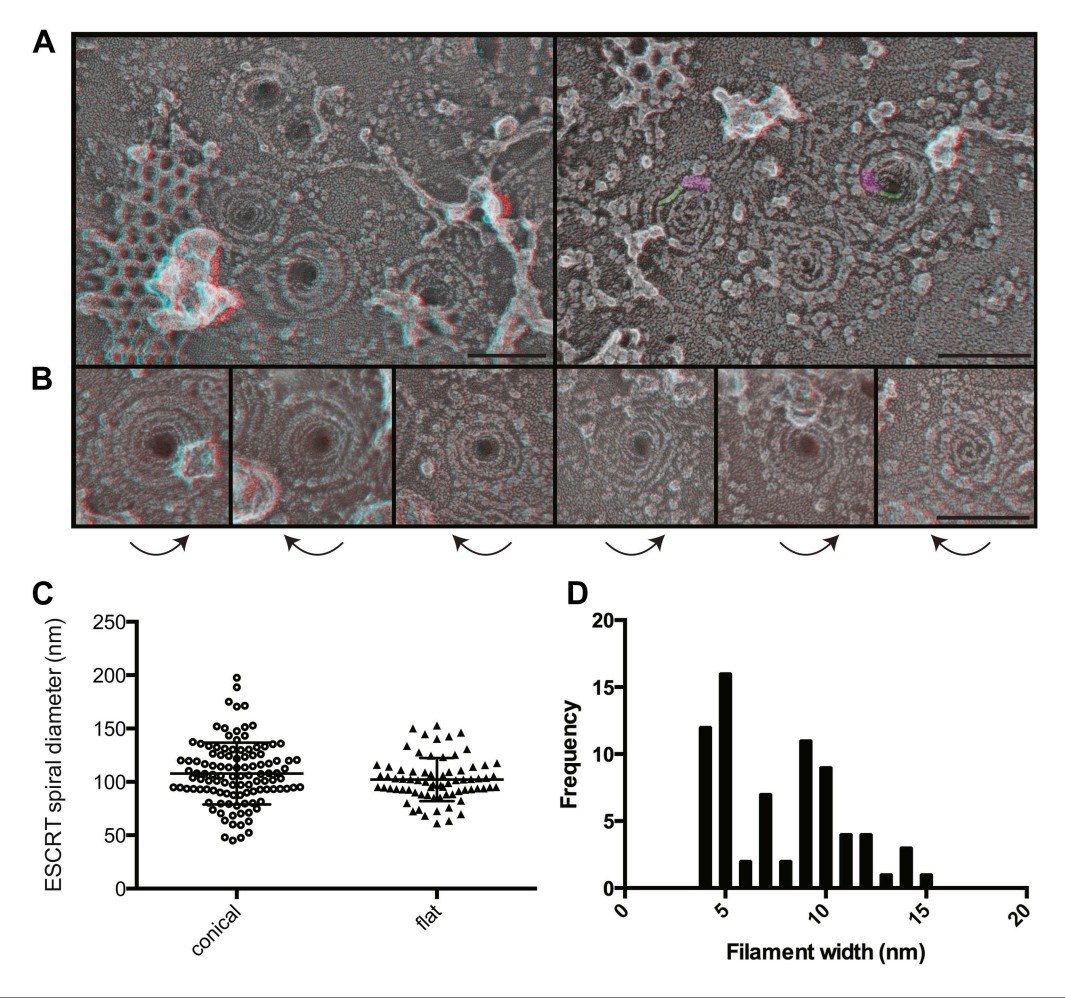

**Figure 3**. Structural characteristics of endogenous ESCRT-III spirals. (**A**) Survey views of filaments and spirals on plasma membranes of HeLa cells depleted of Vps4A & Vps4B show range of shapes, filament diameter, and direction of spiraling. Examples of abrupt changes in filament diameter are highlighted in color (thicker filaments in magenta and thinner filaments in green) in right panel. Use view glasses for 3D structure (left eye = red). Scale bar represents 100 nm. (**B**) Views of individual spirals with direction of spiral from perimeter towards center as shown. Each box corresponds to 185 nm. (**C**) Outer diameter of spirals defined as conical or flat based on appearance in 3D (n = 184, 61% conical 108 ± 29 nm; 39% flat 102 ± 20 nm). (**D**) Distribution of filament widths measured at three points per spiral.

overexpressing specific components of ESCRT-III (*Hanson et al., 2008*; *Bodon et al., 2011*). ESCRT-III assembly must therefore be controlled by something other than the availability of subunits. One possibility is that centripetal growth of a membrane-attached filament determines the diameter of the final assembly. Interestingly, ESCRT-III spirals surround and cover an area of membrane that corresponds well with that needed to generate a vesicle ~50 nm in diameter—the typical size of ILVs in mammalian cells (*Murk et al., 2003*)—suggesting that an initial ESCRT-III circle might define the content of an incipient vesicle.

For insight into how ESCRT-III filaments grow on the membrane, we asked if they preferentially spiral in one direction or the other. Surprisingly, we found both left- and right-handed spirals even in the same field of view (*Figure 3*). This was true both for obviously conical and flat spirals. Even without knowing precisely how subunits are arranged within the filaments, it is clear that changes along the spiral have the potential to affect positioning of functionally important motifs in the ESCRT-III proteins. For example, changes in the recently described N-terminal membrane insertion motif (*Buchkovich et al., 2013*) could contribute to membrane deformation.

The apparent diameter of the finest filaments in ESCRT-III spirals was ~4 nm (*Figure 3D*). Because individual ESCRT-III proteins contain a ~7 nm helical hairpin that is 3–4 nm wide (*Muziol et al., 2006*; *Bajorek et al., 2009*; *Xiao et al., 2009*; *Martinelli et al., 2012*), their assembly into filaments must involve interactions along an approximately longitudinal axis as previously envisioned for filaments and tubes assembled in vitro (*Lata et al., 2008*; *Henne et al., 2012*; *Dobro et al., 2013*; *Effantin et al., 2013*). Filament width varied from 4 to as much as 15 nm in both conical and flat spirals (*Figure 3D*), excluding differences in accessibility to platinum deposition as a primary cause of the differences. Instead, differences in platinum decoration are likely to reveal variations in molecular structure (*Bachmann et al., 1985*). Wider filaments—sometimes with a split down their middle suggesting the presence of parallel substrands—were most prevalent at the perimeter while thin filaments occurred throughout but were almost always present near the spirals' center. Filaments sometimes abruptly changed width (see examples highlighted in *Figure 3A*) further supporting the presence of subfilaments within wider structures. Given the functional requirement for two ESCRT-III nucleating sites in ESCRT-II (*Teis et al., 2010*) or Alix (*Pires et al., 2009*), one possibility is that parallel filaments create the outer turns of a spiral but give way to single filaments with the potential for tighter curvature near the spiral's center. The ability of specific ESCRT-III proteins to cap filament assembly (*Teis et al., 2008*) could promote transition from wide to narrow filaments by ending one of two (or more) parallel filaments. Understanding the functional significance and control of differences in filament content and shape will be an important question for the future.

## Coassembly of CHMP4A with other ESCRT-III proteins creates membrane-deforming filaments

We next wondered what it is about endogenous ESCRT-III filaments that promotes their assembly into stereotyped membrane-deforming spirals. While deep-etch EM does not have the resolution to define the arrangement of individual subunits, it allows us to compare filament shape and organization to that of filaments assembled in transfected cells. We previously found that overexpressed FLAG-tagged CHMP4 proteins create extensive networks of interconnected curved but flat filaments (*Hanson et al., 2008*) and others have seen that CHMP4B and yeast Snf7 behave similarly in vitro (*Pires et al., 2009*; *Henne et al., 2012*). To rule out any effects of the acidic FLAG-tag at the N-terminus of CHMP4A in our earlier study (*Hanson et al., 2008*), we examined filaments formed by untagged CHMP4A (*Figure 4A*). This was important because the FLAG-tag might specifically interfere with a recently described N-terminal membrane insertion motif in CHMP4-family proteins (*Buchkovich et al., 2013*). As before, CHMP4A filaments circle and spiral along the membrane to form an anastomosing network but do not couple this to changes in membrane shape (*Figure 4A*). CHMP4A filaments also remain flat when built from a constitutively 'open' CHMP4(α1–α5) truncation mutant (*Shim et al., 2007*; *Figure 4B*) or a K52E mutant that enhances Snf7 assembly in yeast (*Henne et al., 2012*) (not shown). These consistently flat networks indicate that interface(s) needed to deform the membrane are either not present or not exposed in CHMP4 homopolymers even when intramolecular autoinhibitory contacts are released.

Given that ESCRT-III function in cellular events requires more than one ESCRT-III protein (*Babst et al., 2002*; *Morita et al., 2010, 2011*), we hypothesized that coassembly of CHMP4 with additional ESCRT-III proteins might be required to create filaments with the shape of the endogenous filaments above. To test this idea, we examined the effect of coexpressing CHMP4A with CHMP2A based on the fact that the ESCRT-III function in a wide variety of pathways and organisms requires members of both the CHMP4 and CHMP2 subfamilies of proteins (*Babst et al., 2002*; *Morita et al., 2010, 2011*). Previous biochemical analyses showed that full-length CHMP4A and CHMP2A do not efficiently coassemble while heteropolymers form readily when either protein is activated by deleting its autoinhibitory domain (*Shim et al., 2007*). Strikingly, membranes from cells co-expressing 'open' CHMP4A(α1–α5) and full-length CHMP2A were studded with eversions lined by spiraling filaments (*Figure 4C*). Corresponding eversions were apparent protruding from the tops of whole cells (*Figure 4D*); underlying filaments were revealed by delipidating fixed samples with detergent (*Figure 4E*). Similar membrane-deforming filaments developed when we coexpressed full-length CHMP4A with 'open' CHMP2A(α1–α5) (not shown), demonstrating that the shape changes do not depend on C-terminal autoinhibitory sequences in either protein. The fact that ESCRT-III heteropolymers differ from the sum of their parts supports a model in which subunits such as CHMP2A coassemble with CHMP4 to change the structure of the filament and its effects on membrane shape. Similar changes in filament morphology were previously seen with purified proteins in vitro (*Henne et al., 2012*) suggesting that distinctive

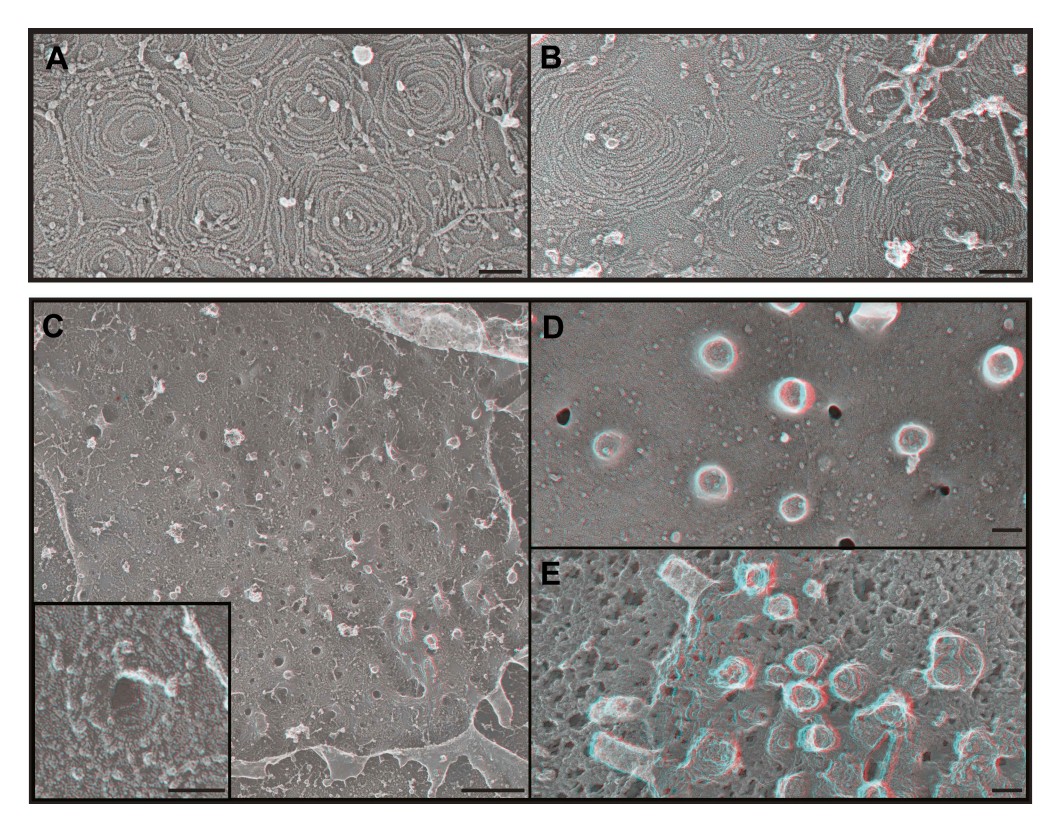

**Figure 4**. ESCRT-III polymer structure in transfected cells. Anaglyphs of plasma membranes from COS-7 cells expressing (**A**) untagged CHMP4A, (**B**) CHMP4A(α1–α5), and (**C–E**) CHMP4A(α1–α5) and full-length CHMP2A together. (**C**) Cytoplasmic surface of the plasma membrane from an unroofed cell with high magnification shown in the inset, (**D**) corresponding top of a whole cell, and (**E**) top of a whole cell extracted with Triton X-100 and saponin after fixation to expose the underlying protein scaffold. Use view glasses for 3D structure (left eye = red). Scale bars represent 100 nm except in **C** where the scale bar on the survey view corresponds to 500 nm.

heteropolymer shape may be a general feature of ESCRT-III biology. An important question for the future will be to define how recruitment and assembly of appropriate ESCRT-III subunits is regulated to create the requisite polymer shape.

## ESCRT-III filaments surround nascent HIV-1 Gag assemblies

A challenge in thinking about the function of ESCRT-III spirals trapped by depleting Vps4 (*Figures 2 and 3*) is that they are typically not in contact with nascent or released vesicles. If spirals represent 'scars' left behind after vesicle budding, they have separated from the vesicles they helped to create. An alternative possibility is that spirals represent the preferred arrangement of ESCRT-III when it assembles spontaneously as a 'blank' in the absence of cargo (and Vps4). Efforts to see ESCRT-III spirals on the surface of endosomes where their function in vesicle biogenesis is better defined were complicated by poor retention and accessibility of these organelles to deep-etch EM as well as the development of stacked class E compartments similar to those seen in *Saccharomyces cerevisiae* (*Nickerson et al., 2006*, and not shown). We therefore turned to the best-characterized ESCRT-dependent event at the plasma membrane, Human Immunodeficiency Virus (HIV-1) budding. HIV-1 assembly is driven by polymerization of the virally encoded Gag polyprotein, which recruits cellular ESCRT proteins to facilitate virion release from the plasma membrane and can be expressed alone to produce virus-like particles (VLPs) (*Gheysen et al., 1989*; *Karacostas et al., 1989*; *Sundquist and Krausslich, 2012*). Current thinking based on electron tomography of immature virions (*Wright et al., 2007*; *Carlson et al., 2008*; *Briggs et al., 2009*) and bud sites (*Carlson et al., 2008*) is that ESCRTs and especially ESCRT-III play important roles both in completing the viral sphere (that is only 2/3 covered

by polymerized Gag) and in severing its connection to the cell. ESCRT-III and Vps4 are transiently recruited to Gag assemblies to mediate release (*Jouvenet et al., 2011*). This machinery is typically thought to act on the cytoplasmic surface of the plasma membrane to constrict the vesicle neck and release a viral particle (*Sundquist and Krausslich, 2012*), although a recent study using fluorescently labeled proteins and superresolution imaging raised the possibility of similar constriction from within the viral particle (*Van Engelenburg et al., 2014*). Using deep-etch EM allows us to capture snapshots of this process while assessing the relationship between ESCRT-III and HIV-1 Gag as a readily recognizable cargo.

HEK293T cells transiently expressing HIV-1 Gag (*Figure 5*) or Gag-GFP (*Figure 5—figure supplement 1*) produce abundant VLPs that are readily apparent by deep-etch EM both on and around cells as well as beneath unroofed plasma membranes (*Figure 5A–C*). Release of VLPs was corroborated by fluorescence microscopy of cells expressing Gag-GFP (*Figure 5—figure supplement 1*) and by isolation and immunoblotting of VLPs (not shown). Unroofed plasma membranes display unique circular and semi-spherical protein assemblies ranging in size up to the diameter of VLPs that appear to be nascent Gag assemblies (*Figure 5D*). In order to ascertain that these in fact contain Gag, we immunodecorated unroofed cells with an antibody specific to the membrane-proximal matrix (MA) domain of Gag (*Figure 5E–H*). Gold particles were numerous around putative Gag assemblies on unroofed plasma

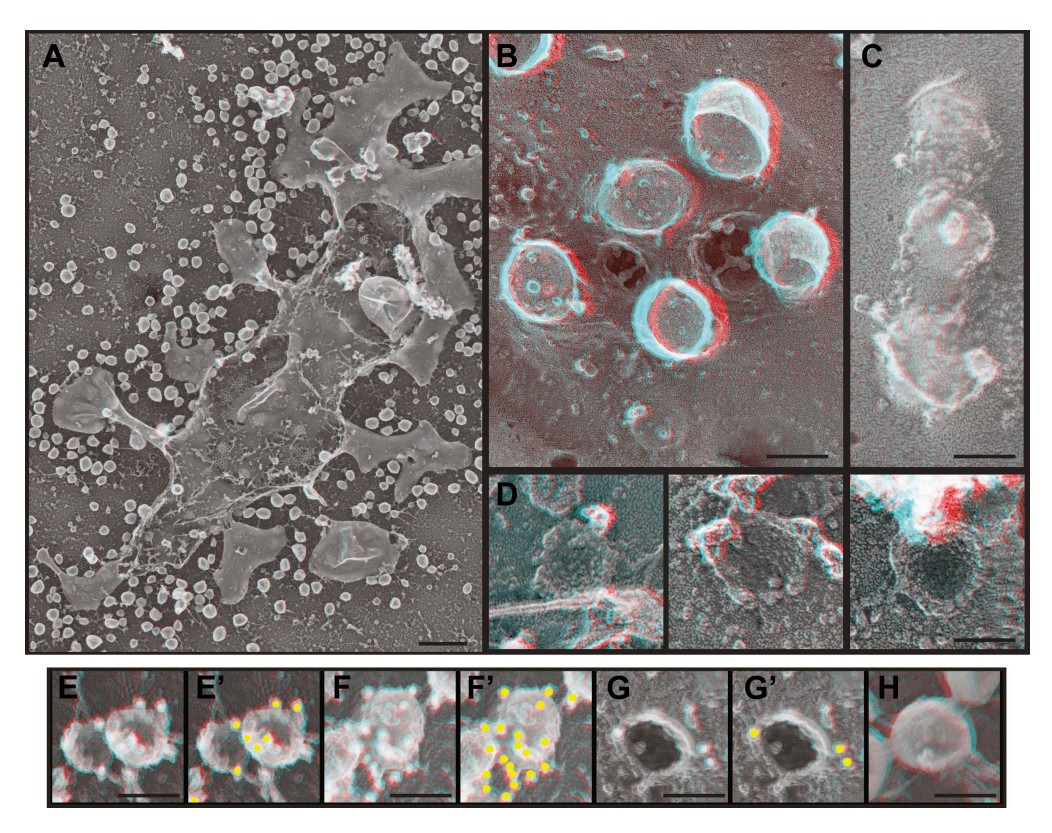

**Figure 5**. Deep-etch EM of HIV-1 VLP budding. (**A**) Low magnification view of an unroofed HIV-1 Gag-transfected HEK293T cell surrounded by VLPs. (**B**) Top view of whole cell budding VLPs. (**C**) View of unroofed plasma membrane showing bumps corresponding to VLPs trapped underneath the membrane. (**D**) Views of unroofed plasma membrane showing Gag assemblies exposed on the cytoplasmic surface of the plasma membrane. (**E–H**) Immunodecoration of Gag on detergent extracted plasma membranes (**E** and **E'**), detergent extracted VLPs (**F** and **F'**), intact unroofed plasma membranes (**G** and **G'**) and intact VLPs (**H**). (**E'**, **F'** and **G'** are same as **E**, **F** and **G** but show gold in yellow). Use view glasses for 3D structure (left eye = red). Scale bars represent (**A**) 500 nm, (**B–H**) 100 nm.
The following figure supplements are available for figure 5:

**Figure supplement 1**. VLP formation by HIV-1 Gag-GFP.

membranes (*Figure 5E,E'*) and around VLPs (*Figure 5F,F'*) when samples were delipidated by detergent extraction after fixation. When membranes were intact, immunodecoration of Gag assemblies was limited to their perimeter (*Figure 5G,G'*) and was abolished in released VLPs (*Figure 5H*) as expected. By deep-etch EM, Gag-GFP assemblies were less uniform in size and shape than those containing Gag (*Figure 5—figure supplement 1C*), consistent with the irregular distribution of Gag-GFP seen by thin section EM (*Pornillos et al., 2003*) and with the decreased Gag content of VLPs containing Gag fused to similarly sized fluorescent proteins (*Gunzenhauser et al., 2012*). Notably there was no evidence by direct viewing or immunolabeling (not shown) to indicate the presence of ESCRT-III on or near any of these Gag assemblies. This is not surprising given live cell studies showing that ESCRT-III and Vps4 are only transiently recruited after Gag assembly is essentially complete (*Baumgartel et al., 2011*; *Jouvenet et al., 2011*).

To explore the role of ESCRT-III filaments in VLP biogenesis, we therefore once again depleted cells of Vps4 to stabilize ESCRT-III in its assembled state. As expected, expressing a dominant negative mutant of Vps4A (Vps4A E228Q) or silencing Vps4 as above increased the amount of Gag-GFP (*Figure 6—figure supplement 1A*) or Gag (*Figure 6—figure supplement 1B*) on the plasma membrane and decreased release of VLPs as detected by particle analysis methods (data not shown). Strikingly, in cells lacking Vps4 Gag assemblies on the plasma membrane were now often surrounded by a single filament not seen in control cells (*Figure 6*). The encircling filaments appeared similar to the ESCRT-III filaments examined above, each ranging from 4.5–12.8 nm (average 8.6 ± 1.6 nm, n = 96) in width with the majority wide enough to contain more than one ~4 nm substrand. Immunolabeling confirmed that ESCRT-III proteins localized to the perimeter of Gag assemblies coincident with the filaments (*Figure 6D*). Interestingly, the encircling filaments surrounded Gag assemblies of various sizes ranging from ~60–150 nm suggesting that the threshold amount of Gag needed to activate ESCRT-III assembly may be decreased in the setting of reduced Vps4 leading to premature nucleation of the ESCRT-III ring. Alternatively, these smaller Gag assemblies may indicate a role for ESCRT-III filaments in confining Gag (or other cargo proteins) in a domain that normally expands upon remodeling by Vps4. More deeply invaginated Gag assemblies were sometimes surrounded by what appeared to be tightly spiraled ESCRT-III filaments, and immunolabeling confirmed that ESCRT-III was present on the cytoplasmic rim and surrounding surface of these invaginating VLPs (*Figure 6D*, right panel and not shown). These spirals may correspond to the darkly stained ring previously noted in thin section EM analysis of HIV-1 budding from cells depleted of CHMP2A (and therefore unable to engage Vps4) (*Morita et al., 2011*) and seem likely to represent trapped intermediates in the biogenesis of VLPs.

How do Gag assemblies recruit ESCRT-III filaments? We noticed that ESCRT-III filaments (in cells depleted of Vps4) were frequently connected to the central Gag assembly by perpendicular 'struts' ~14 nm long (*Figure 6*, see green highlighting in inset). These struts resemble previous negative stain images of purified Alix bound perpendicularly to Snf7 filaments (*Pires et al., 2009*) although based on size they could also correspond to ESCRT-I (*Boura et al., 2011*), an ESCRT-I-ESCRT-II supercomplex (*Boura et al., 2012*), or something else altogether. The C-terminal p6 domain of Gag contains a PTAP motif (essential for interaction with the ESCRT-I protein Tsg101 [*Garrus et al., 2001*]) and a LYPX$_n$L motif (involved in interaction with ALIX [*Strack et al., 2003*]) and has been shown to be essential for budding of Gag VLPs as well as HIV-1. To determine whether these motifs are required for ESCRT-III recruitment, we deleted the p6 domain from Gag (referred to as GagΔp6). In cells expressing GagΔp6, very few VLPs were released and instead large numbers of arrested Gag-containing buds accumulated on the surface (*Figure 7* and not shown). Importantly, this was true regardless of whether Vps4 was present or not, confirming that the p6 domain is required for VLP release. On unroofed plasma membranes Gag assemblies similar to those formed by full-length Gag were observed but were not surrounded by ESCRT-III rings (*Figure 7*), demonstrating that p6 sequences are required to recruit ESCRT-III. Furthermore, no struts were apparent, consistent with the hypothesis that these correspond to ESCRT-I and/or Alix. We were, however, unable to confirm the identity of the struts because available antibodies were unsuitable. Detailed mutagenesis coupled with antibodies appropriate for immunodecoration on these samples will be needed to conclusively identify these structures.

## Models for ESCRT-III function in vesicle biogenesis

The data presented here provide the first views of endogenous ESCRT-III filaments forming the rings and spirals frequently drawn in models of ESCRT function but not previously seen on cellular membranes. Earlier studies of various ESCRT-III polymers in vitro and in transfected cells inspired thinking

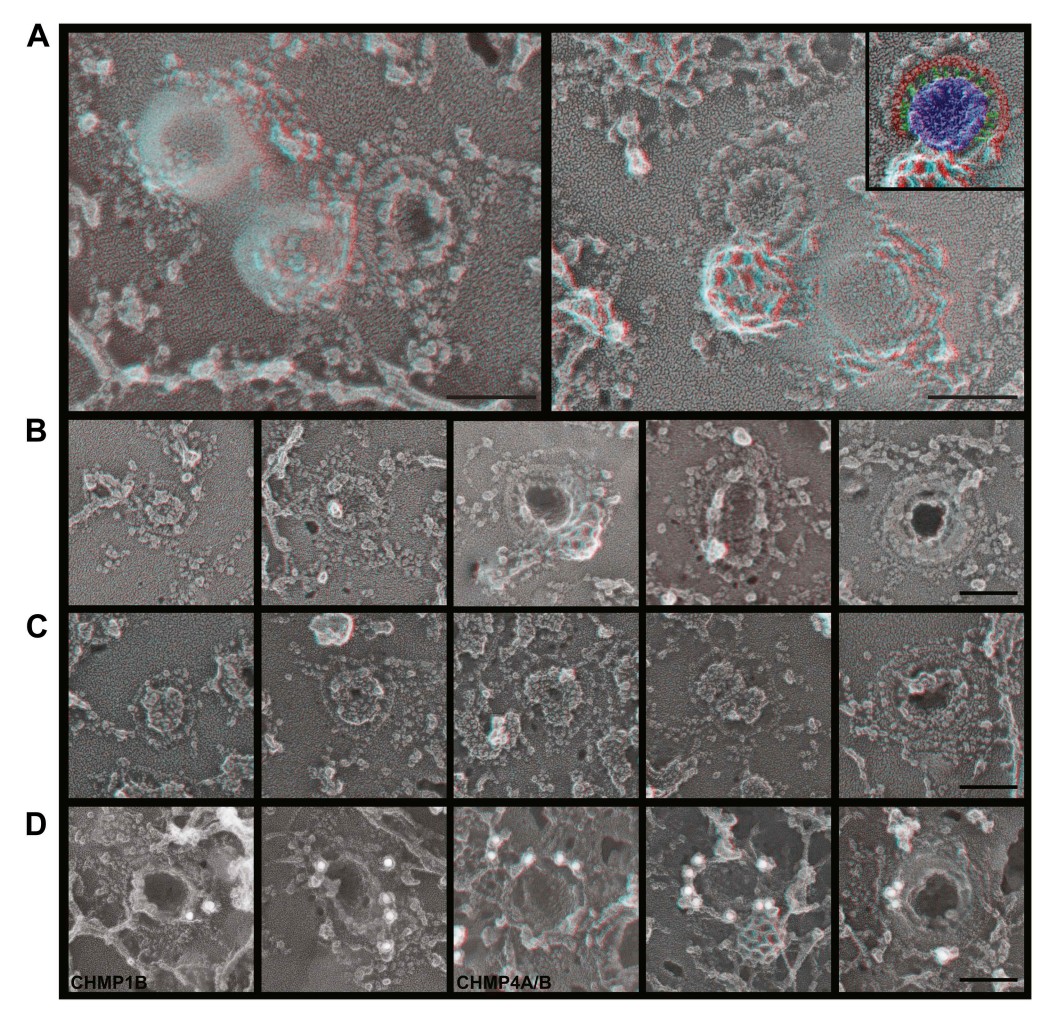

**Figure 6**. ESCRT-III filaments surround Gag assemblies in cells depleted of Vps4. (**A**) HEK293T cells treated with Vps4A & Vps4B siRNA accumulate unique filament-encircled Gag assemblies. Inset is pseudo-colored to show the central Gag assembly (blue), surrounding filament (red), and perpendicular 'struts' between them (green). Each field also shows the occasionally seen subplasmalemmal VLP bump to provide a sense of scale. (**B**) Individual views of Gag assemblies, (**C**) Gag-GFP assemblies, and (**D**) Gag assemblies immunodecorated with indicated gold conjugated antibodies. Scale bars represent 100 nm. Each box in **B**–**D** is a 320 nm square. Use view glasses for 3D structure (left eye = red).

The following figure supplements are available for figure 6:

**Figure supplement 1**. VLP release is impaired by inactivating or depleting Vps4.

about how different filamentous and tubular structures might function in vesicle biogenesis and membrane scission. Ideas ranged from purse-string like constriction of a ring (*Saksena et al., 2009*) to polymer-driven membrane buckling (*Lenz et al., 2009*) to the more prevalent dome-based membrane scission model (*Fabrikant et al., 2009*) and its variants (*Boura et al., 2012*; *Elia et al., 2012*; *Henne et al., 2012*). We propose a modified model that builds on many of these previous ideas, interpolating between the rings of ESCRT-III we see around HIV-1 Gag assemblies and the completed conical spirals that accumulate on the membrane (*Figure 8*). In this model, a first role for ESCRT-III is to encircle and thereby delineate cargo and membrane destined for inclusion in a vesicle or VLP (*Figure 8A*). The encircling filament develops when nucleating factors activated by cargo form struts around a central domain and in turn recruit ESCRT-III. Either because of intrinsic filament curvature or because of multiple contacts with cargo, ESCRT-III filaments grow to surround the cargo (*Figure 8A*). Because

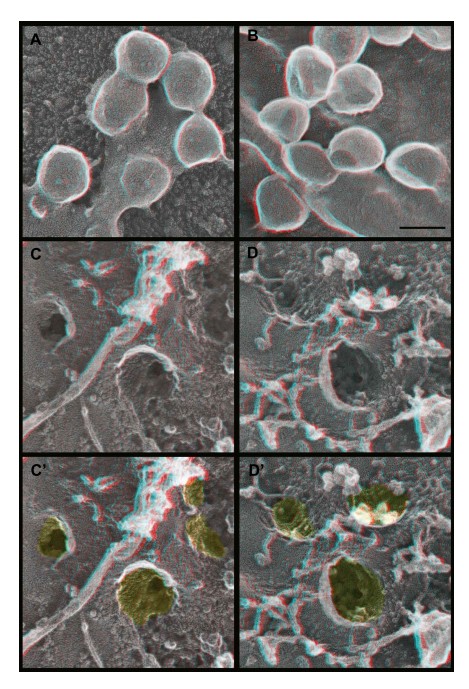

**Figure 7**. Deep-etch EM of Gag budding without its ESCRT-recruiting p6 domain. (**A**) Top surface of HEK293T cell expressing GagΔp6 showing blocked VLP budding. (**B**) Top surface of Vps4-depleted HEK293T cell expressing GagΔp6 showing similar blocked VLP budding. (**C**) Unroofed plasma membrane corresponding to **A**. (**D**) Unroofed plasma membrane corresponding to **B**. Note the invaginated Gag assemblies with no surrounding ESCRT-III ring (**C** and **D**). **C'** and **D'** are identical to **C** and **D** but with Gag assemblies colored yellow for clarity. Scale bar represents 100 nm. Use view glasses for 3D structure (left eye = red).

encircled Gag assemblies uniquely accumulate when Vps4 is missing (**Figure 6**), we propose that an important and previously unappreciated role for Vps4 is to release connection(s) between the ESCRT-III filament and cargo, perhaps displacing the struts while also opening the ring. The ESCRT-III filament can then grow to adopt its preferred shape as a conical spiral (**Figure 8B**) potentially driving neck constriction and vesicle release as shown (**Figure 8C**). Finally (and not shown), Vps4 is also responsible for disassembling ESCRT-III filaments to recycle their subunits during or after vesicle release.

While this model remains speculative because it is based on static images of ESCRT-III polymers stabilized by depleting Vps4, it both supports important aspects of earlier models describing ESCRT-III driven membrane remodeling (**Fabrikant et al., 2009**; **Wollert et al., 2009**; **Elia et al., 2012**; **Henne et al., 2012**) and provides explanations for a few previously enigmatic observations. Interestingly, there are similarities between the different forms of ESCRT-III seen here and structures previously described by EM of purified yeast proteins (**Henne et al., 2012**). Specifically, Henne et al. found that combining ESCRT-II as a nucleating factor with ESCRT-III proteins (Vps20 and Snf7 or a combination of four core ESCRT-III proteins) generated rings of ESCRT-III ~50–70 nm in diameter, appropriate for surrounding a domain needed to create 25–35 nm ILVs characteristic of yeast MVBs. In our images, the ~110 nm average outer diameter (**Figure 3C**) of ESCRT-III spirals would in an analogous manner support formation of the ~50 nm vesicles typical of mammalian ILVs and extracellular vesicles. Henne et al. also found that combining ESCRT-III proteins with each other

in the absence of ESCRT-II created spring-like three-dimensional coils. Not clear was whether (and how) rings might convert into spiraling filaments. We suggest that Vps4 may have a general role in converting a cargo confining ring to a vesicle-generating spiral, likely by separating ESCRT-III from nucleating factors seen as struts in our model. The idea that Vps4 plays specific and essential roles both before and after membrane scission is supported by a number of observations. Fluorescently tagged Vps4 is recruited to growing viral particles before they are released (**Baumgartel et al., 2011**; **Jouvenet et al., 2011**), Vps4 appears at two stages prior to cellular abscission (**Elia et al., 2011**), ESCRT-III and Vps4 are recruited simultaneously to the centroid of dividing *Crenarchaea* (**Samson et al., 2011**), and perturbing Vps4 function and/or regulation changes ILV size and/or release in yeast (**Nickerson et al., 2010**; **Wemmer et al., 2011**; **Adell et al., 2014**).

A prediction of our model is that interfering with Vps4 activity—whether by depleting it or expressing a dominant negative mutant—will trap ESCRT-III around the base of incipient vesicular structures and in particular around the base of emerging Gag assemblies. This trapped ESCRT-III will remain attached to Gag inside any VLPs that are ultimately released. This in turn will increase ESCRT-III recovered in whatever VLPs are released, something that we (data not shown) and others (**Van Engelenburg et al., 2014**) have in fact observed.

While we favor the model shown in **Figure 8**, one can also envision other ways in which the remodeling of ESCRT-III rings and spirals could contribute to vesicle formation and membrane scission. Future coupling of the ability to see membrane-bound ESCRT assemblies using deep-etch EM with

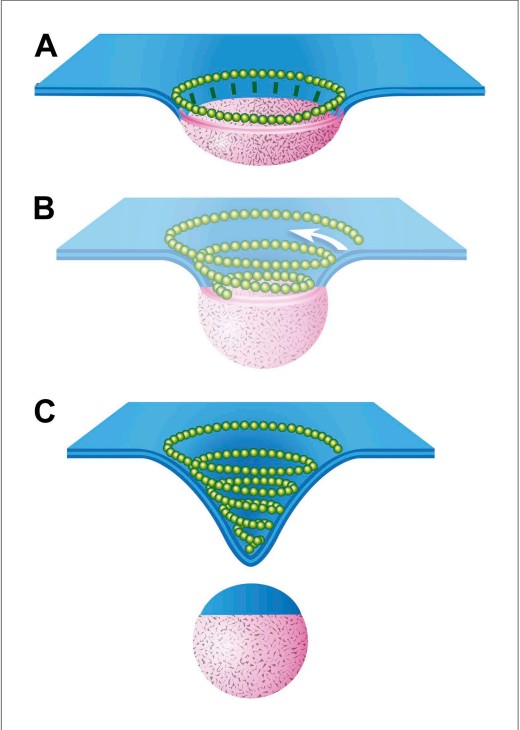

**Figure 8**. Speculative model describing ESCRT-III and Vps4 function in vesicle biogenesis and release. (**A**) Cargo - loosely defined to include either HIV-1 Gag or material destined for incorporation into ILVs–is concentrated in the circular domain shown in pink. After reaching some threshold, cargo recruits and/or activates factors to initiate ESCRT-III assembly. These are represented here by green 'struts' perpendicular to the cargo perimeter. Once nucleated, the ESCRT-III filament extends to surround and confine cargo. In the absence of Vps4, this intermediate accumulates. (**B**) When present, we propose a new role for Vps4 in which it is engaged to break connection(s) between ESCRT-III and cargo, thereby allowing the ESCRT-III spiral to grow into its preferred spiral shape. ESCRT-III recruits new membrane into the neck as it grows, shown by the addition of blue membrane to the budding vesicle. (**C**) A fully assembled ESCRT-III spiral narrows the membrane neck, ultimately driving vesicle release. DOI: 10.7554/eLife.02184.013

techniques capable of resolving and potentially controlling protein and membrane dynamics will be important in gaining new insight into these processes. Defining the detailed temporal relationship between Vps4 activity and ESCRT-III assembly during virus or vesicle formation will be particularly important. Additionally, while we have taken advantage of the fact that ESCRT polymers are present on the plasma membrane to facilitate deep-etch EM imaging, future studies of their function at the plasma membrane are clearly warranted. Overall, the work described here establishes that endogenous ESCRT-III takes the form of filaments that form circles and conical spirals on the membrane. We propose that these rings and spirals, while longer-lived than normal, represent functional states of a general ESCRT-III membrane scission machine.

## Materials and methods

### Cell culture

HeLa, HEK293T, U2OS, and COS-7 cells originally derived from ATCC were grown in DME (Invitrogen, Grand Island, NY) containing 10% fetal bovine serum (Atlanta Biologics, Atlanta GA) and 2 mM L-glutamine.

### Plasmids and antibodies

pCMV55 encoding HIV-1 Gag was kindly provided by Dr Lee Ratner (Washington University, St. Louis MO) and used as previously described (*Shim et al., 2007*). The p6 domain of Gag was deleted by introducing a stop codon to remove 52 amino acids from the C-terminus and introduce an XhoI site using the primer AAAAAACTCGAGTTAAAAA TTCCCTGGCCTTCCCTTG.

Full length and p6 deleted Gag were cloned into pcDNA4TO (Invitrogen). pGag-EGFP was obtained through the NIH AIDS Reagent Program, Division of AIDS, NIAID, NIH (Cat#11468) from Dr Marilyn Resh (*Hermida-Matsumoto and Resh, 2000*). Plasmids encoding ESCRT-III proteins have been previously described, including untagged human CHMP4A (also referred to as hSnf7-1) in pcDNA3 (*Lin et al., 2005*), CHMP4A(α1–α5) in pcDNA3.1FLAG (*Shim et al., 2007*), and CHMP2A full length and α1–α5 truncation in pcDNA3.1FLAG (*Shim et al., 2007*).

Antibodies used include rabbit polyclonals against Gag (MA-specific; kind gift from Dr Lee Ratner, Washington University School of Medicine, St. Louis MO), CHMP6 (PA5-21831; ThermoScientific, Waltham MA), CHMP4A (*Lin et al., 2005*), CHMP4B (*Shiels et al., 2007*), CHMP2A (sc67227; Santa Cruz, Dallas, TX), CHMP2B (ab33174; AbCam, Cambridge, England), CHMP1B (14639-1-AP; Protein Tech Group, Chicago, IL), VPS4B (*Lin et al., 2005*) and mouse monoclonals against tubulin (Sigma, St. Louis MO) and α-SNAP (SySy, Gottingen, Germany).

siRNA duplexes targeting VPS4A (CCGAGAAGCUGAAGGAUUAdTdT) and VPS4B (CCAAAGA AGCACUGAAAGAdTdT) have been previously described (*Kieffer et al., 2008*) and were from ThermoScientific.

## Polymer sedimentation analysis

Detergent solubility assays to monitor ESCRT-III polymer assembly after Vps4 depletion were as previously described (*Shim et al., 2007*).

## Immunofluorescence

HeLa and HEK293T cells plated on glass coverslips were stained as previously described (*Shim et al., 2007*). Confocal imaging was performed on an Olympus FV500 or FV1200 microscope using a 60x 1.4 NA objective. Maximum intensity projections were prepared using ImageJ (version 1.47u). Brightness and contrast were adjusted as necessary with Adobe Photoshop (Adobe Systems, San Jose, CA) and composite figures were prepared in Adobe Illustrator.

## Transfections and sample preparation for deep-etch EM

Cells plated at ~70% confluence were transfected with 15 nM each of siRNA duplexes targeting VPS4A and VPS4B using Dharmafect#1 according to manufacturer guidelines. Cells were trypsinized ~24 hr later and replated onto 12 mm poly-L- or poly-D-lysine coated BioCoat coverslips (BD Biosciences, East Rutherford NJ). Plasma membranes were prepared the following day, typically 40–50 hr after initial siRNA transfection. This was done as previously described (*Hanson et al., 2008*). Briefly, coverslips were washed in 30 mM Hepes, pH 7.4, 100 mM NaCl, 2 mM CaCl2 and then dipped into an intracellular buffer (30 mM Hepes, pH 7.2, 70 mM KCl, 5 mM MgCl2, and 3 mM EGTA) and subjected to a brief pulse of ultrasound before transfer into the same buffer containing fixative (2% glutaraldehyde or 2% PFA if immunostaining was planned). The area of coverslip with the highest yield of plasma membranes was identified by phase contrast microscopy and trimmed with a diamond knife to ~3 × 3 mm.

Transfection of Gag or Gag-GFP encoding plasmids was performed using Lipofectamine 2000 (Life Technologies, Grand Island, NY) according to the manufacturer's instructions. When combined with siRNA transfections, plasmid transfections were carried out 24 hr after initial introduction of siRNA. ESCRT-III encoding plasmids were transfected as previously described (*Hanson et al., 2008*) again using Lipofectamine 2000 (Life Technologies). To extract fixed samples with detergent (*Figures 4E, 5E,F*), fixed coverslips were incubated for 2 hr in buffer containing 1% Triton X-100 and 0.1% saponin.

## Immunogold antibody decoration

Antibody staining was performed as previously described (*Hanson et al., 2008*) using 18 nm gold-conjugated goat anti-rabbit or anti-mouse (Jackson Immunoresearch, West Grove, PA) antibodies.

## Freezing, replicating, and imaging deep-etch samples

Samples were prepared essentially as described (*Hanson et al., 2008*) except that platinum (~2 nm, as before) was evaporated onto samples from 17–18° above the horizontal. Replicas were viewed on a JEOL 1400 transmission electron microscope at two different tilt angles (either ±5 or ±10°) and images were captured using an AMT camera. Digital image pairs were made into anaglyphs by converting one each to red and blue/green, layering them on top of each other using the screen blending mode in Adobe Photoshop, and finally aligning them to each other using the auto-align function. Composite figures were prepared using Adobe Illustrator. Digital measurements were made using ImageJ (v. 1.47u).

## Acknowledgements

We thank members of the Hanson laboratory and our colleagues at Washington University for helpful discussions and Erin Blumer for help drawing the model in *Figure 8*. This work was supported by R01 GM076686 from the National Institutes of Health to PIH.

## Additional information

### Funding

| Funder | Grant reference number | Author |
| --- | --- | --- |
| National Institutes of Health | R01 GM076686 | Anil G Cashikar, Soomin Shim, Robyn Roth, Michael R Maldazys, Phyllis I Hanson |

The funders had no role in study design, data collection and interpretation, or the decision to submit the work for publication.

## Author contributions

AGC, PIH, Conception and design, Acquisition of data, Analysis and interpretation of data, Drafting or revising the article; SS, RR, JEH, Conception and design, Acquisition of data, Analysis and interpretation of data; MRM, Acquisition of data, Analysis and interpretation of data

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
