## [Decision Letter]

Thank you for sending your work entitled “Deep-etch EM of cellular ESCRT-III reveals intrinsic potential for membrane remodeling and relationship to HIV-1 budding” for consideration at *eLife*. Your article has been favorably evaluated by a Senior editor, a Reviewing editor, and 3 reviewers, one of whom, Heinrich Gottlinger, has agreed to reveal his identity.

The Reviewing editor and the reviewer discussed their comments before we reached this decision, and the Reviewing editor has assembled the following comments to help you prepare a revised submission.

The authors describe deep etch EM imaging of ESCRT-III assemblies at the plasma membrane, both in the presence and absence of HIV-1 Gag, which assembles into virus-like particles (VLPs). VPS4A/B-depleted cells are used to prevent rapid ESCRT-III filament disassembly and to arrest VLP budding. The study makes four important claims: 1) Conical spiraling filaments of ESCRT-III proteins can be observed at the plasma membranes of cells lacking Vps4 proteins. 2) Spiraling ESCRT filaments assemble at the necks of nascent VLPs. This observation supports the dominant model that ESCRT-III filaments mediate membrane scission at the bud neck, but is the first compelling visualization of those filaments. 3) The ESCRT filaments act from the cytosolic face of the bud. This configuration has been assumed in other models, but there has been little direct evidence for this topology and the idea is not universally accepted. This is therefore also an important observation. Indeed, the authors probably understate the importance of this observation and it should be discussed more thoroughly and in light of the recent Science paper from the Lippincott-Schwartz lab). 4) Co-overexpression of activated CHMP4A and CHMP2A proteins induces the extrusion of membrane tubes on the plasma membrane, implying that CHMP2A provides extrusion activity to CHMP4A. All of these observations appear to be consistent with the dome model for membrane constriction and fission, at least in general terms. Although this is already the leading model in the field, there has been little direct experimental evidence to support or refute it and the contributions are therefore important and warrant publication in *eLife*. As detailed below, however, there are still a series of key scientific and presentation issues that need to be address prior to publication.

Major issues:

1) The array of quite different Gag/VLP images (e.g., as seen in Figure 6) is potentially very informative and may be capturing different stages of the budding process. It seems surprising, however, that “early-arrested“ budding structures (e.g., as apparently seen in Figure 6, right panel) are observed when Vps4 is depleted. One might have expected to see primarily “late-arrested” budding structures in which Gag is up inside the virion and the bud neck is ringed by ESCRT-III filaments (e.g., as apparently seen in Figure 6, far right panel). Do the authors think they are observing terminally arrested states or a range of different intermediate states? What is the range and variability of the structures seen? Can this variability be quantified or the sites classified? Why do the authors think the HIV budding structures appear so variable, and why are they not all roughly the same size?

A related issue is that it is difficult to follow how (and why) the authors think the different structures that they see are linked mechanistically, including the flat spirals, relatively flat Gag patches encircled by ESCRT-III rings, and hole-lined spirals. How do the these structures relate to one another? Which are expected to convert into others? Which might dead-end intermediates due to the lack of Vps4? What role might the apparent thickening of the filaments towards the outside of the spirals imply (and why is this feature missing from the final model)? What are the mechanistic implications of the apparent lack of preferred spiral directionality? How do all of these observations fit into a coherent model for budding?

Because some of the mechanistic interpretations will inevitably be speculative, it is at least critically important to establish with certainty that the different polymers have been correctly assigned. Two experiments are required. A) The authors should dually and differentially immunolabel Gag/CHMP complexes in the same sample (to confirm that the central, flat patches really are Gag assemblies and that the encircling rings really are ESCRT-III filaments). B) The authors should image the assemblies formed by Gag proteins that lack the TSG101 (PTAP) and YPXL (ALIX) binding sites. This experiment is expected to produce Gag assemblies lacking ESCRT rings and would greatly strengthen the claim that the encircling rings are indeed ESCRT assemblies. We acknowledge that these experiments may be time consuming, but they are important and seem to be within the expertise of the authors.

2) The images in Figure 6—figure supplement 1 need to be improved (and/or supplemented with thin sectioned-EM images). As it stands, it is difficult to say with confidence that the image shows “many nascent VLPs” attached to the cell or to understand the homogeneity of the nascent viral particles in the absence of Vps4. Most of the membrane blebs don't look like HIV and this could instead simply be a dying cell.

Additional issues (for the authors' consideration):

1) The observed filaments are repeatedly referred to as “functional”. However, many different filamentous and tubular structures have been visualized by different workers, and it is difficult to know whether the filaments visualized here are more functionally relevant than any of the tubes or other types of structures seen previously. An alternative explanation is that the structures seen here represent overgrowths due to unchecked ESCRT-III polymerization in the absence of Vps4 activity. We therefore recommend discussing this issue and removing the term “functional” throughout and replacing it with a more nuanced discussion placing these structures in context with others. A related issue is that the authors should consider whether Vps4 functions might include: A) detaching ESCRT components from “cargo” (as suggested, although strictly speaking, is “cargo” is the right term for Gag?), and B) preventing the formation of the overgrown structures that are being imaged here.

2) It is frequently hard to relate the statements made in the text to features in the images. For example, Figure 6 is at the heart of the paper, but it is difficult to follow. We recommend adding scale bars in Figure 6, arrows to highlight which densities are attributed to ESCRT-III filaments, which correspond to ALIX/ESCRT-I “struts”, and which to Gag, together with clearer explanations for the basis of the assignments, the reproducibility of the observed structures, and the stage of budding to which the authors believe image corresponds. Perhaps this would all best be clarified with accompanying diagrams and/or three-dimensional re-renderings of the structures with the ESCRTs and Gag colored differently. These illustrations could replace Figure 8, which is a generic model that is similar to what has appeared in many review articles. Other features that should be clarified better include: A) Figure 4, the transverse breaks in the platinum every 4 nm (and the reliability of this measurement?), and precisely where split filaments can be seen in this or other images, B) In Figure 3–figure supplement 1, the evidence for the statement “immunodecoration inside broken but not fully unroofed Vps4-depleted cells showed that ESCRT-III containing eversions were also present along the cell's top surface”, and C) Consider adding additional annotations on at least some of the figures, e.g., marking gold beads, caveolae, clathrin assemblies, etc., to help orient viewers.

3) There is significant redundancy with the 2008 JCB paper from the same authors. For example, the flat spirals formed by CHMP4 overexpression shown in Figure 7 are similar to those shown in Figure 4 of the 2008 paper, as are the eversions. Figure. 7 could therefore be condensed. Similarly, Figure 3 is a useful control showing that the filaments are CHMPs, but the immunolabeling data in Figure 6 are more relevant (and actually quite impressive), and in the light of these data (and also in light of the relatedness of Figure 2 to similar observations in the 2008 JBC paper), the authors should also consider contracting/combining Figure 2 and Figure 3.

4) As the authors point out, it is surprising that there would not be a preferred directionality to the ESCRT-III spirals, and the directionality assignments in Figure 4 are not particularly convincing. Figure 4 is a minor part of the paper and could be removed without affecting the main conclusions. Alternatively, the assigned filament paths should be explicitly overlaid on the EM images (and please clarify whether the handedness assignments are relative to the “in” or “out” direction).

5) Filament diameters may be exaggerated due to the Pt stain. This complicates comparison to filaments imaged by other methods and this issue should be noted in the discussion.

6) The rationale for using Gag-GFP in some of the experiments is unclear.

---

## [Author Response]

*1) The array of quite different Gag/VLP images (e.g., as seen in*
Figure 6*) is potentially very informative and may be capturing different stages of the budding process. It seems surprising, however, that “early-arrested” budding structures (e.g., as apparently seen in*
Figure 6*, right panel) are observed when Vps4 is depleted. One might have expected to see primarily “late-arrested” budding structures in which Gag is up inside the virion and the bud neck is ringed by ESCRT-III filaments (e.g., as apparently seen in*
Figure 6*, far right panel). Do the authors think they are observing terminally arrested states or a range of different intermediate states? What is the range and variability of the structures seen? Can this variability be quantified or the sites classified? Why do the authors think the HIV budding structures appear so variable*, *and why are they not all roughly the same size?*

We agree that the extent of variability in the Gag/VLP assemblies is somewhat surprising, although probably not entirely unexpected based on the variability in the size of Gag assemblies quantified in cells transiently expressing fluorescently tagged Gag proteins (Gunzenhauser *et al.*, 2012). We have added a sentence describing the range of diameters that we observe for Gag assemblies, but do not think that quantifying their relative frequency makes sense because cell-to-cell variability in Gag expression is significant in transiently transfected cells and may affect the concentration and distribution of the resulting Gag assemblies. We cannot directly quantify this relationship using unroofed plasma membranes because we do not know how much Gag was present in the intact cell. To circumvent this problem, we attempted to make tetracycline-inducible stable Gag-expressing cell lines so that every cell would express approximately the same amount. Three separate attempts to generate these were unsuccessful, apparently because the amount of Gag expressed did not reach the threshold needed to assemble VLPs.

In the setting of reduced Vps4 activity, we were surprised by the fact that both smaller and larger Gag assemblies recruited ESCRT-III rings (Figure 6). We now explicitly discuss the significance of this observation, suggesting as possible explanations that (a) ESCRT-III filaments might be recruited prematurely to small assemblies because of reduced Vps4 activity, (b) ESCRT-III and Vps4 might regulate assembly and growth of cargo-containing domains, or (c) these structures are part of the normal continuum of nascent VLPs and appear as flat as they do because of the geometry of the bottom plasma membrane to which they are attached. All of these are possible, especially given the fact that our new studies of GagΔp6 (Figure 7 and not shown) reveal assembly of deeper “late arrested” budding structures when ESCRT-III is not recruited.

*A related issue is that it is difficult to follow how (and why) the authors think the different structures that they see are linked mechanistically, including the flat spirals, relatively flat Gag patches encircled by ESCRT-III rings*, *and hole-lined spirals. How do the these structures relate to one another? Which are expected to convert into others? Which might dead-end intermediates due to the lack of Vps4? What role might the apparent thickening of the filaments towards the outside of the spirals imply (and why is this feature missing from the final model)? What are the mechanistic implications of the apparent lack of preferred spiral directionality? How do all of these observations fit into a coherent model for budding?*

The reviewers are of course correct that the connections we suggest among different static images are by definition speculative, but we think they are supported by a number of considerations both in our work and that published by others. We have rewritten significant parts of the text in an effort to better convey our thinking, and hope that the reviewers will agree that our ideas are now more accessible.

Specific additions in response to the above questions include:

While the apparent “early” or small Gag assemblies encircled by ESCRT-III were unexpected and therefore noteworthy, we also see (and show) that depleting Vps4 traps what appear to be later arrested structures in which invaginated Gag assemblies are surrounded by filament spirals (selected panels in Figure 6). These contributed to our thinking about connections between states, as represented in the model in Figure 8.

While we see clear variations in filament width, we do not yet know whether and if so how this is functionally significant and mention explicitly the need to address this in the future. We speculate on some interesting possibilities (wider filaments prevalent at the perimeter might have less effect on membrane curvature than the thinner ones, they might curve with a preferred diameter, they might recruit different binding partners, etc.), and propose that filament capping might be responsible for the transition between widths. We left differences in filament width out of our final model both because we do not yet know how significant these are and because we see occasional spirals that are entirely comprised of thin filaments.

We also do not know how significant the observed lack of directionality is, but have consistently found filaments spiraling in both directions over the course of many experiments examining ESCRT-III spirals in Vps4-depleted cells. These studies clearly establish that once directionality in a spiral is established it persists.

*Because some of the mechanistic interpretations will inevitably be speculative, it is at least critically important to establish with certainty that the different polymers have been correctly assigned. Two experiments are required. A) The authors should dually and differentially immunolabel Gag/CHMP complexes in the same sample (to confirm that the central, flat patches really are Gag assemblies and that the encircling rings really are ESCRT-III filaments). B) The authors should image the assemblies formed by Gag proteins that lack the TSG101 (PTAP) and YPXL (ALIX) binding sites. This experiment is expected to produce Gag assemblies lacking ESCRT rings and would greatly strengthen the claim that the encircling rings are indeed ESCRT assemblies. We acknowledge that these experiments may be time consuming, but they are important and seem to be within the expertise of the authors*.

A) Immunolabeling. We fully agree with the importance of confirming the identity of the different polymers. Convincing dual labeling on deep-etch replicas (e.g. using 18 vs. 12 nm gold) is challenging under the best of circumstances and turned out to be problematic in this case because of limited access to epitopes in Gag recognized by antibodies we tried. We have therefore instead included images of Gag immunodecoration to clearly identify Gag polymers as such (Figure 5); this complements ESCRT-III immunodecoration around the same type of structures (Figure 6). As we used a Gag-MA specific antibody, labeling on native membranes was mostly limited to the perimeter of the Gag assembly under native conditions while it appeared around the entire assembly after detergent extraction.

B) Gag deletion. We have added images of GagΔp6 budding (new Figure 7), and as expected see no recruitment of struts or encircling ESCRT-III filaments.

*2) The images in*
Figure 6—figure supplement 1
*need to be improved (and/or supplemented with thin sectioned-EM images). As it stands, it is difficult to say with confidence that the image shows “many nascent VLPs” attached to the cell or to understand the homogeneity of the nascent viral particles in the absence of Vps4. Most of the membrane blebs don't look like HIV and this could instead simply be a dying cell*.

The original image was chosen because it showed what we thought was a particularly nice example of a cell in which the central area was unroofed to expose the plasma membrane while areas at the periphery remained intact to show nascent VLPs. We however agree with the reviewer that the attached VLPs were not as homogeneous as typically seen, and we have therefore replaced it with an image of a whole cell expressing Gag (and depleted of Vps4) that clearly shows the “many nascent VLPs” we referred to. In addition we have also included fluorescence microscopy images of GagGFP in cells with blocked (Vps4EQ) or reduced (siRNA) Vps4 function.

*Additional issues (for the authors*' *consideration):*

*1) The observed filaments are repeatedly referred to as “functional”. However, many different filamentous and tubular structures have been visualized by different workers, and it is difficult to know whether the filaments visualized here are more functionally relevant than any of the tubes or other types of structures seen previously. An alternative explanation is that the structures seen here represent overgrowths due to unchecked ESCRT-III polymerization in the absence of Vps4 activity. We therefore recommend discussing this issue and removing the term “functional” throughout and replacing it with a more nuanced discussion placing these structures in context with others. A related issue is that the authors should consider whether Vps4 functions might include: A) detaching ESCRT components from “cargo” (as suggested, although strictly speaking, is “cargo” is the right term for Gag?), and B) preventing the formation of the overgrown structures that are being imaged here*.

We agree that a considered discussion of what we can and cannot learn from looking at polymers trapped by depleting Vps4 is important, and have added relevant sentences to the manuscript. We specifically consider possible roles for pre-scission remodeling by Vps4 (see Figure 8). We have also removed the term “functional” throughout (except in the last sentence of the paper where it is used in a speculative context). We want to emphasize that our original use of the term functional was motivated by the desire to distinguish the polymers studied here (that contain only endogenous proteins) from those in earlier studies of either overexpressed or purified proteins.

*2) It is frequently hard to relate the statements made in the text to features in the images. For example,*
Figure 6
*is at the heart of the paper, but it is difficult to follow. We recommend adding scale bars in*
Figure 6*, arrows to highlight which densities are attributed to ESCRT-III filaments, which correspond to ALIX/ESCRT-I “struts”, and which to Gag, together with clearer explanations for the basis of the assignments, the reproducibility of the observed structures, and the stage of budding to which the authors believe image corresponds. Perhaps this would all best be clarified with accompanying diagrams and/or three-dimensional re-renderings of the structures with the ESCRTs and Gag colored differently. These illustrations could replace*
Figure 8*, which is a generic model that is similar to what has appeared in many review articles. Other features that should be clarified better include: A)*
Figure 4*, the transverse breaks in the platinum every 4 nm (and the reliability of this measurement?), and precisely where split filaments can be seen in this or other images, B) In Figure 3–figure supplement 1, the evidence for the statement “immunodecoration inside broken but not fully unroofed Vps4-depleted cells showed that ESCRT-III containing eversions were also present along the cell's top surface”, and C) Consider adding additional annotations on at least some of the figures, e.g., marking gold beads, caveolae, clathrin assemblies, etc., to help orient viewers*.

We appreciate the suggestions, and have enhanced and clarified connections between text and figures throughout. We have added color to draw attention to particular features and some additional scale bars. We are keeping the model in Figure 8 because it represents what we see and plays a critical role in conveying why and how we think the different structures may be interconnected. It is indeed consistent with some previous models, but this is not a weak point. It also will play an important role in motivating future experimental tests of these ideas (for example, a clear prediction of this model is that ESCRT-III filaments should grow after Vps4 acts on an encircling filament; this is something that can be tested in future live cell or *in vitro* imaging studies).

Addressing other features in need of clarification:

A) While transverse breaks in the platinum along ESCRT-III filaments at ∼4 nm intervals are reproducible (and distinct from the 5.5 nm breaks seen along actin), they are a result of platinum decoration and may or may not directly correlate to a structural feature of the underlying filament. They are also subtle (and require high-magnification viewing of optimal replicas). We therefore decided to remove this sentence.

B) We have added an inset to the figure in question to highlight the gold particles present under the surface of the plasma membrane (Figure 2—figure supplement 1, formerly Figure 3–figure supplement 1).

C) We have added some color to help orient viewers without detracting from how data is presented in 3D anaglyphs.

*3) There is significant redundancy with the 2008 JCB paper from the same authors. For example, the flat spirals formed by CHMP4 overexpression shown in*
Figure 7
*are similar to those shown in*
Figure 4
*of the 2008 paper, as are the eversions. Figure. 7 could therefore be condensed. Similarly,*
Figure 3
*is a useful control showing that the filaments are CHMPs, but the immunolabeling data in*
Figure 6
*are more relevant (and actually quite impressive), and in the light of these data (and also in light of the relatedness of*
Figure 2
*to similar observations in the 2008 JBC paper), the authors should also consider contracting/combining*
Figure 2
*and*
Figure 3.

We have reorganized the manuscript and added explanations for the transfected cell experiments shown: (former Figure 7, now Figure 4), to clarify how these differ from what was shown in the 2008 paper. These results are important in understanding the contributions of different ESCRT-III proteins to ESCRT-III filament morphology and build on but are not redundant with our earlier paper.

The immunolabeling of the spirals (former Figure 3, now Figure 2) is important in its own right and also establishes that the antibodies used in Figure 6 work. We agree that combining the former Figure 2 and Figure 3 into one makes sense, and have done this.

*4) As the authors point out, it is surprising that there would not be a preferred directionality to the ESCRT-III spirals, and the directionality assignments in*
Figure 4
*are not particularly convincing.*
Figure 4
*is a minor part of the paper and could be removed without affecting the main conclusions. Alternatively, the assigned filament paths should be explicitly overlaid on the EM images (and please clarify whether the handedness assignments are relative to the “in” or “out” direction)*.

We think this figure (now Figure 3) provides important characterization of the endogenous ESCRT-III filaments and the spirals they form. The lack of directionality is an important feature even if we do not yet understand its implications. We now define arrow direction in the legend.

*5) Filament diameters may be exaggerated due to the Pt stain. This complicates comparison to filaments imaged by other methods and this issue should be noted in the discussion*.

We have added a sentence describing the limitations of platinum shadowing. Filament diameters may or may not be exaggerated depending on the degree of decoration vs. shadowing.

*6) The rationale for using Gag-GFP in some of the experiments is unclear*.

We originally used Gag-GFP to enable tracking of VLPs by fluorescence and include the EM-derived images of encircled Gag-GFP as additional support for our identification of encircled Gag assemblies. As expected based on previous work, Gag-GFP assemblies are less regular than those formed by Gag alone (see Figure 5—figure supplement 1).